# Sensorimotor Network Segregation Predicts Long-Term Learning of Writing Skills in Parkinson’s Disease

**DOI:** 10.3390/brainsci14040376

**Published:** 2024-04-12

**Authors:** Nicholas D’Cruz, Joni De Vleeschhauwer, Martina Putzolu, Evelien Nackaerts, Moran Gilat, Alice Nieuwboer

**Affiliations:** 1Research Group for Neurorehabilitation (eNRGy), Department of Rehabilitation Sciences, KU Leuven, Tervuursevest 101, Box 1500, B-3001 Leuven, Belgium; nicholas.dcruz@kuleuven.be (N.D.); joni.devleeschhauwer@kuleuven.be (J.D.V.); evelien.nackaerts@kuleuven.be (E.N.); moran.gilat@kuleuven.be (M.G.); 2Department of Experimental Medicine (DIMES), Section of Human Physiology, University of Genoa, 16132 Genoa, Italy; martina.putzolu@unige.it

**Keywords:** Parkinson’s disease, motor learning, rehabilitation, micrographia, resting-state fMRI, network segregation, prediction model

## Abstract

The prediction of motor learning in Parkinson’s disease (PD) is vastly understudied. Here, we investigated which clinical and neural factors predict better long-term gains after an intensive 6-week motor learning program to ameliorate micrographia. We computed a composite score of learning through principal component analysis, reflecting better writing accuracy on a tablet in single and dual task conditions. Three endpoints were studied—acquisition (pre- to post-training), retention (post-training to 6-week follow-up), and overall learning (acquisition plus retention). Baseline writing, clinical characteristics, as well as resting-state network segregation were used as predictors. We included 28 patients with PD (13 freezers and 15 non-freezers), with an average disease duration of 7 (±3.9) years. We found that worse baseline writing accuracy predicted larger gains for acquisition and overall learning. After correcting for baseline writing accuracy, we found female sex to predict better acquisition, and shorter disease duration to help retention. Additionally, absence of FOG, less severe motor symptoms, female sex, better unimanual dexterity, and better sensorimotor network segregation impacted overall learning positively. Importantly, three factors were retained in a multivariable model predicting overall learning, namely baseline accuracy, female sex, and sensorimotor network segregation. Besides the room to improve and female sex, sensorimotor network segregation seems to be a valuable measure to predict long-term motor learning potential in PD.

## 1. Introduction

People with Parkinson’s disease (PD) experience a variety of motor and non-motor symptoms [1] caused primarily, but not exclusively, by dopamine deficiency in the basal ganglia, affecting the sensorimotor striatum. The sensorimotor striatum is also implicated in the acquisition and retention of motor skills [2,3,4], resulting in an impaired motor learning ability in PD [5,6,7]. As a result, PD patients require more neural activity compared to healthy controls to reach similar levels of motor performance and show an overreliance on attentional processing during acquisition and retrieval of learned patterns [8,9].

Motor learning is an indispensable part of motor rehabilitation and can be defined as a process in which motor skills are acquired and consolidated following practice and time [6].

Although micrographia is an early and common symptom in PD, only a few studies have investigated interventions to alleviate this debilitating symptom, characterized by a stable or progressive reduction in writing size [10]. Real-time writing size can be improved using external cueing, such as visual target lines and auditory rhythms [11,12]. Interestingly, a recent review showed that handwriting-specific training interventions, including the addition of cueing, resulted in immediate improvements [13]. Additionally, previous results from our lab did not only show immediate effects, but also consolidation and retention of motor learning after a six-week intensive writing training program aimed at improving micrographia [14].

Motor learning is thus possible in PD patients, although it requires greater effort and more exposure compared to healthy individuals [5,9]. Importantly, certain patient subgroups showed different training responses, highlighting the need for predictors of motor learning to optimize and individualize training in PD [15]. Clinical markers, such as the presence of freezing of gait (FOG), are associated with worse motor learning [15,16,17]. Similarly, baseline performance and cognitive ability may predict response to training [18,19]. However, besides clinical phenotyping, neural markers of motor learning, such as functional brain organization, may provide additional relevant information to improve prediction of training improvements in PD [20].

Resting-state functional Magnetic Resonance Imaging (RS-fMRI) is a useful tool to analyze functional brain organization by capturing the degree of functional connectivity (FC) within and between resting-state networks [21]. Several studies demonstrated that training can alter RS-FC, both in healthy individuals and PD patients, indicating the sensitivity of RS-FC to motor learning processes [22,23,24]. In the present cohort, we recently showed that FC within the dorsal attention network increased in the group that received the writing training [25]. Nevertheless, functional brain organization as a predictor of motor learning has not yet been investigated in PD.

One important metric of functional brain organization is the modularity or segregation of a network, reflecting the degree of within-network to between-network FC. Higher segregation across all networks at rest, as well as of visuomotor networks with increasing task complexity, has been associated with better performance of a bimanual coordination task in healthy young and older adults [26,27]. Moreover, in terms of learning, higher sensorimotor within-network RS-FC and lower visual–sensorimotor between-network connectivity, both indicative of higher sensorimotor segregation, predicted better motor learning outcomes in healthy young adults [20,28].

To the best of our knowledge, no studies have investigated the predictive value of RS-FC brain organization for motor learning capacity in PD, or its added value to clinical predictors. Here, we studied the association between baseline clinical measures and brain network segregation, and improvements in writing performance after writing training in PD patients, thereby contributing to the development of individualized rehabilitation. Based on the work discussed above, we expected higher segregation of the sensorimotor network and lower baseline performance to predict larger improvements in writing performance.

## 2. Materials and Methods

### 2.1. Participants

This study constituted a secondary analysis of previously published studies [14,15]. Thirty-two PD patients with RS-fMRI scans who received intensive writing training in a large, randomized placebo-controlled study were included in this analysis. All participants were diagnosed with idiopathic PD by a neurologist according to the United Kingdom PD Society Brain bank criteria [29]. The inclusion criteria consisted of (1) right-handedness, measured by the Edinburgh Handedness Inventory [30]; (2) Hoehn and Yahr (H&Y) stage I–III [31]; (3) the presence of handwriting difficulties, indicated by a score ≥ 1 on item II.7 of the Movement Disorder Society Unified Parkinson’s Disease Rating Scale (MDS-UPDRS) [32]. The exclusion criteria were (1) a Mini-Mental State Examination (MMSE) score < 24 [33]; (2) other upper limb impairments that interfered with writing performance; (3) self-reported visual impairments including color blindness, and (4) contra-indications for MRI. The study was approved by the local Ethics Committee of the University Hospitals Leuven (S54132) in accordance with the Declaration of Helsinki (version 1967). Prior to participation, all participants signed an informed consent form after detailed explanation of the protocol.

### 2.2. Study Design

All participants performed a home-based largely unsupervised 6-week writing training program, except for a weekly visit from a researcher who managed training progression. Training was performed 5 days per week, 30 min per day, while optimally medicated. To assess the effects of training, all patients were tested at three time points, i.e., pre-training, post-training and 6-week follow-up. Tests were performed ON medication, approximately 1 h after medication intake, which was standardized across the three time points. The baseline session included tests examining clinical characteristics, as well as tests assessing writing performance. Furthermore, MRI data were collected at baseline for all participants during the ON medication state. This scanning session included the acquisition of a high-resolution anatomical image, a resting-state functional scan, a magnetic susceptibility fieldmap and task-based functional scans, consecutively. The resting-state scan was collected before the task-based scan, as well as before the behavioral assessment to avoid any influence of task performance on resting-state brain organization. Findings with respect to the task-based scans are reported elsewhere, and as such were not considered here [34,35,36].

### 2.3. Experimental Procedure

Clinical tests consisted of the motor part of the MDS-UPDRS (MDS-UPDRS-III) and the H&Y scale. Cognitive function was assessed using the Montreal Cognitive Assessment (MOCA) [37], and affective symptoms with the Hospital Anxiety and Depression Scale (HADS) [38]. Levodopa equivalent daily dosage (LEDD) was calculated and medication was kept constant throughout the study [39,40]. Using the New Freezing of Gait questionnaire [41], patients were classified as freezers or non-freezers, based on whether they had experienced FOG in the past month. Sleep complaints were assessed using items 7 and 8 from the MDS-UPDRS part I, and unimanual dexterity was assessed with the Purdue Pegboard Test number of pegs for the right hand [42].

Writing performance was assessed outside the scanner at the three time points using two tasks: a trained task, and an untrained task. To assess acquisition and retention of the learnt task rather than transfer of learning, we focused on the trained task, which comprised a three-loop sequence with visual target zones [14]. The task was performed without and with a cognitive dual task (counting low or high tones) to interrogate the ability to withstand interference. The task was performed in two sizes, i.e., 0.6 (small) and 1.0 (large) cm, visually indicated by the distance between the bottom of the blue and top of the yellow zone (Figure 1). All participants performed three runs of 27 s trials on a touch-sensitive writing tablet with a sampling frequency of 200 Hz and a spatial resolution of 32.5 μm.

The six-week writing training aimed to improve writing performance, more specifically writing accuracy in relation to the desired size, and is described in detail elsewhere [14]. Briefly, it consisted of exercises on paper as well as on a touch-sensitive tablet in the presence of visual target zones. The difficulty of writing exercises was gradually increased over the course of training and dual tasking was also introduced to facilitate consolidation by asking patients to count tones during the task. Compliance with training was recorded through self-reported logbooks and was calculated as a percentage of the required training dose. Training progressions were followed up by one of the researchers.

### 2.4. Writing Data Processing and Learning Outcomes

Data from the touch-sensitive tablet, collected after the MRI scan, were filtered at 7 Hz with a 4th-order Butterworth filter [43] and processed using MATLAB (R2011, The Mathworks Ltd., Natick, MA, USA). The differences between the local minima and maxima of the individual strokes were calculated to determine the writing amplitude (in % of target size). In addition, the coefficient of variation of amplitude (COVampl) was determined as well as the writing speed (cm/s). Further, the deviation from the target was calculated as the absolute difference between the target amplitude and the achieved amplitude (i.e., Deviation = |100 − writing amplitude|). A principal component analysis (PCA) was performed on the data of the large size condition in single and dual tasks over the three time points to determine the main components (explaining > 80% variance) of writing performance and to reduce dimensionality. Only the large size was used as it presented greater difficulty at pre-training (smaller amplitudes relative to the target) [14] and therefore greater scope for learning. Varimax rotation of the components was performed, and component regression scores were saved. Component scores were averaged for single and dual tasks within the time points to obtain a more robust measure of writing performance. Robustness was ensured as similar performance on both tasks would not impact the scores while tradeoff between the two tasks (better performance in one relative to the other) would cancel out, warranting a stable estimate of writing performance. Difference scores were then computed across time points to characterize acquisition of learning (pre- to post-6 weeks of training), retention of learning (post-training to 6-week follow-up without training), and overall learning (pre-training to follow-up, 12 weeks).

### 2.5. Neuroimaging Data

#### 2.5.1. Acquisition Parameters

MRI scanning was carried out using a Philips Achieva 3T scanner (Best, The Netherlands). A standard 32-channel head coil was used with foam cushions to restrict head motion. High-resolution T1-weighted anatomical scans [T1 turbo field echo (TFE) sequence, duration = 383 ms; slice number = 182; slice thickness = 1.2 mm; repetition time (TR) = 9.624 s; echo time (TE) = 4.6 ms; flip angle = 8°; matrix = 256 × 256; FOV = 218.4 × 250 × 250 mm] were acquired for each participant. Functional resting-state data were acquired with T2*-weighted functional images using gradient echo-planar imaging (EPI) pulse sequences (duration = 435 s; slice number = 31; slice thickness = 4 mm; TR = 1700 ms; TE = 33 ms; flip angle = 90°; matrix = 64 × 64; FOV = 230 × 124 × 230 mm; voxel size = 3.59 × 3.74 × 4 mm). During the resting-state scan, participants were asked to keep their eyes open and look at the white cross on the black screen. Moreover, a gradient echo fieldmap was obtained (duration = 234 s; slice number = 35; slice thickness = 4 mm; TR = 750 ms; TE = 5.76 ms; flip angle = 90°; matrix = 96 × 96; FOV = 192 × 192 × 147 mm).

#### 2.5.2. Preprocessing

Preprocessing of imaging data was performed using fMRIPrep 1.5.9 [44] (RRID:SCR_016216), which is based on Nipype 1.4.2 [45] (RRID:SCR_002502). A brief description of the most important steps follows. For a detailed overview of all preprocessing steps, please see the Appendix A.

The T1-weighted anatomical images were segmented into cerebrospinal fluid (CSF), gray matter (GM) and white matter (WM) and normalized to the Montreal Neurological Institute (MNI) standard space using a non-linear transformation. Resting-state functional images underwent motion realignment, slice timing correction and unwarping and co-registration to the anatomical image using a gradient-echo fieldmap. Functional images were unwarped and normalized to the MNI standard space in a single transformation. Noise components were calculated using ICA-AROMA and regressed from functional scans after spatial smoothing with an 8 mm FWHM gaussian kernel. Further denoising was performed using the CONN toolbox (version 19.c) [46] for MATLAB (version R2019b) and SPM12 (https://www.fil.ion.ucl.ac.uk/spm/software/spm12/, accessed on 16 June 2020). Functional volumes were scrubbed if the framewise displacement exceeded 0.5 mm or if the mean BOLD signal change was an outlier (i.e., it exceeded the third quartile and was 1.5 times the interquartile range). Average timeseries extracted from GM, WM and CSF masks, motion realignment parameters, their derivatives, and the quadratic effects (24 parameters) as well as scrubbing dummy variables were included as regressors. Moreover, linear detrending and temporal high-pass filtering (0.008 Hz-Inf) were applied.

#### 2.5.3. Quality Control

Based on previous literature, at least 4 min of RS-fMRI data is required to reliably estimate functional connectivity [47]. Hence, participants with more than 108 scrubbed volumes (based on a TR of 1.7 s) were excluded, resulting in a total of 28 participants in this study (13 freezers, 15 non-freezers). All imaging data were also visually inspected for gross artefacts.

### 2.6. Functional Connectivity Analysis and Outcomes

To identify the networks in the brain, the Cole-Anticevic Brain-wide Network Partition version 1.1 [48] was used. This neurobiologically principled atlas combines 718 regions of interest (ROIs) covering the entire cortex, subcortex and cerebellum into 12 networks. The network names and spatial extent are shown in Figure 2. Cortical ROIs were created in subject space by surface reconstruction and parcellation based on the Glasser Multimodal atlas [49] performed in FreeSurfer (version 6.0), and then moved to standard volumetric space. Subcortical and cerebellar ROIs were defined using the Cole-Anticevic subcortical template in standard volumetric space.

Average timeseries for each ROI were extracted and Pearson’s correlation test was performed between these timeseries. Correlation values were subsequently transformed using the Fisher r-to-Z transformation to obtain a 718 × 718 symmetrical matrix. ROIs were then sorted within networks and average within- and between-network connectivity was calculated for each network to obtain a 12 × 12 symmetrical matrix. Moreover, negatively correlated connections between regions were excluded, as the value of anti-correlations is controversial [50,51]. Finally, for each network, segregation was calculated as the ratio of the difference between within-network and between-network connectivity to the within-network connectivity [52]. Higher scores on this ratio signify more segregation, which is a marker of efficient neural processing.

### 2.7. Statistical Analysis

Demographic and clinical predictors were selected based on previous literature [15,18,53,54,55,56] and consisted of age, gender, MDS-UPDRS-III, disease duration, Purdue Pegboard test, LEDD, HADS, sleep complaints, FOG presence and MOCA. Improvements in writing over time were analyzed with a repeated measurement analysis of variance with time as a within-subject condition. Post hoc pairwise comparisons were Bonferroni corrected. Associations between baseline writing performance and writing improvements (difference scores) were assessed with Pearson’s correlation. Associations between baseline clinical and neural predictors and writing improvements were assessed using Pearson’s partial correlations, accounting for the influence of baseline writing performance. Confidence intervals (95%) for all estimates were obtained from the bias-corrected and accelerated bootstrap (BCa) using 1000 resamples with replacement (BCa 95%CI). The BCa method for calculating the bootstrap confidence interval accounts for bias and skewness in the bootstrap distribution of the estimate and provides adequate coverage of the true estimate [57,58]. Inference was made based on the bootstrap confidence intervals to ensure that results were not driven by a few influential observations. Predictors with confidence intervals excluding 0 were entered in a multivariable backward linear regression model (criteria for retaining predictors was *p* < 0.1) to obtain the parsimonious combination of significant predictors. All variables were demeaned and scaled to unit variance prior to inclusion in the correlation or regression analyses. Alpha was set at 0.05 and all statistical analyses were performed using SPSS (version 23, SPSS Inc., Chicago, IL, USA).

## 3. Results

The clinical demographics of all 28 PD patients included in the analysis are provided below in Table 1. Participants ranged from early to advanced disease stages but were non-demented. In our sample, patients with FOG had a longer disease duration and higher disease severity scores, as measured with the MDS-UPDRS-III, compared to patients without FOG (Appendix A). Participants complied well with the prescribed training (mean compliance rate: 95.5%, range: 63.3–100%).

### 3.1. Determining Writing Accuracy Outcomes

Principal component analysis including the four outcome measures resulted in the identification of two main components of writing performance (for details, see Appendix A). The first component reflected writing accuracy (57.4% of variance explained), while the second component reflected writing speed (29.5% of variance explained; cumulative: 86.9%). A higher component score implies more accurate and faster writing, respectively. Given that the intensive writing training focused specifically on improving amplitude and accuracy, only the first principal component was included in further analyses. An overview of the changes in this component score over time for all participants is illustrated in Figure 3.

### 3.2. Changes in Writing Accuracy for Acquisition, Retention and Overall Learning, and Associations with Baseline Writing Accuracy

Repeated measures ANOVA revealed a significant within-subject effect of Time (F (2,54) = 8.18, *p* = 0.001, partial eta-square = 0.23), with better writing accuracy at post-training (acquisition; mean difference = 0.66, SD = 1.10, Bonferroni-*p* = 0.012) and follow-up (overall learning; mean difference = 0.44, SD = 0.87, Bonferroni-*p* = 0.037) compared to pre-training. No differences were found between post-training and follow-up (retention; mean difference = −0.22, SD = 0.58, Bonferroni-*p* = 0.176).

Pearson correlation revealed significant negative associations between baseline writing accuracy and gains during acquisition (r = −0.49, BCa 95%CI = −0.62 to −0.34) and overall learning (r = −0.39, BCa 95%CI = −0.54 to −0.18), but not with retention (r = 0.34, BCa 95%CI = −0.1 to 0.65). In other words, participants with lower baseline writing accuracy showed larger gains during acquisition and overall learning. Correlations between acquisition, retention and overall learning revealed that overall learning was strongly driven by acquisition (r = 0.85, BCa 95%CI = 0.60 to 0.96), as opposed to retention (r = −0.12, BCa 95%CI = −0.52 to 0.29). Interestingly, retention was inversely associated with acquisition (r = −0.62, BCa 95%CI = −0.79 to −0.31), suggestive of overlapping and opposing, as well as independent and additive contributions to overall learning.

### 3.3. Clinical and Neural Predictors of Writing Accuracy Improvements

Acquisition of writing accuracy was better in females compared to males (r_partial_ = 0.43, BCa 95%CI = 0.1 to 0.69) but was not associated with any other clinical or neural predictors. Retention of writing accuracy was negatively associated with longer disease duration (r_partial_ = −0.26, BCa 95%CI = −0.53 to −0.01), but no other predictors. Two predictors that were significant in the original sample, namely MDS-UPDRS III for acquisition and sleep complaints for retention, did not perform consistently upon bootstrapping (Table 2), and were thus not included in a multivariable model.

Overall learning gains were larger in non-freezers compared to freezers (r_partial_ = 0.44, BCa 95%CI = 0.06 to 0.74), and in females compared to males (r_partial_ = 0.51, BCa 95%CI = 0.20 to 0.74). Overall learning gains were positively associated with unimanual dexterity (r_partial_ = 0.31, BCa 95%CI = 0.02 to 0.56), sensorimotor network segregation (r_partial_ = 0.57, BCa 95%CI = 0.29 to 0.78), and negatively associated with MDS-UPDRS III scores (r_partial_ = −0.52, BCa 95%CI = −0.79 to −0.08). All significant associations are visualized in Figure 4.

Multivariable backward linear regression was then performed for the overall learning gains in writing accuracy including these five predictors as well as baseline writing accuracy. The final model retained baseline accuracy (beta = −0.51, BCa 95%CI = −0.73 to −0.26), female sex (beta = 0.42, BCa 95%CI = 0.12 to 0.71), and sensorimotor network segregation (beta = 0.49, BCa 95%CI = 0.26 to 0.77) as significant predictors of the overall gains in writing accuracy (model adjusted R^2^ = 0.54, *p* < 0.001).

## 4. Discussion

In this study, we investigated the predictive value of clinical measures and resting-state brain functional network organization for motor learning capacity after writing training in PD patients. On a behavioral level, writing accuracy improved after six weeks of training and was maintained after a follow-up period of six weeks without training in the large writing condition, supporting previous work [14]. In line with our expectations based on previous work [18,19,56], worse baseline writing accuracy was associated with greater learning gains over the short and longer term. After correcting for the influence of baseline writing accuracy, we found female sex to be predictive of better acquisition and shorter disease duration to be predictive of better retention of learning. Moreover, overall gains combining acquisition and retention were positively associated with female sex, absence of FOG, higher unimanual dexterity, and higher sensorimotor network segregation, and negatively associated with motor symptom severity. Importantly, upon combining these predictors in a multivariable model, only baseline accuracy, female sex, and sensorimotor network segregation were predictive of overall learning gains. These findings shed light on the clinical and brain functional determinants of motor skill learning and highlight the complementarity of these indicators to predict motor learning responses in PD.

### 4.1. Clinical Predictors of Motor Learning after Accounting for Baseline Accuracy

We tested several clinical measures including age, sex, cognition, anxiety, depression, sleep complaints, unimanual dexterity and disease severity metrics as possible predictors of writing training. After correcting for baseline writing performance, only sex, unimanual dexterity and disease severity measures (disease duration, motor severity and FOG presence) showed associations with writing gains across any time points. Cognition has often been shown to predict learning following training [19,54,59] and hence the lack of association here is surprising. However, this sample showed relatively preserved cognitive capacity, and the task itself was minimally cognitively demanding, reducing the likelihood of cognition being a driving or limiting factor for learning. In addition, anxiety and depression were not identified as predictors for writing training success. Previous studies have, however, showed a significant impact on training compliance, one of the key factors for training success in PD [60,61]. As such, the high compliance rates in our study may explain why these neuropsychiatric symptoms were not identified as predictors. Nevertheless, future studies should examine the influence of compliance, using objective measures, and neuropsychiatric symptoms on training success in a large sample of PD patients.

On the other hand, female sex was an unexpected positive finding, showing consistent predictive values for acquisition as well as overall gains in writing accuracy. There is no documented effect of sex at birth on motor learning in general, and the advantage may be task-specific. Females on average exhibit superior language abilities, particularly for writing with reports of medium effect sizes [62] and show more efficient (lesser activity and more segregated) neural processing while writing [63]. As such, studies investigating learning in language-related tasks should take sex into account.

Interestingly, fewer sleep complaints measured by MDS-UPDRS items I.7 and I.8 were associated with better retention of learning in the original sample, but this result did not hold upon bootstrapping. This is an ordinal and subjective metric and better objective outcomes of sleep quality could be obtained. Indeed, previous work in young adults, quantifying sleep quality using actigraphy, found that time spent awake after sleep onset was negatively associated with subsequent learning of a motor sequencing task [64]. Future studies should consider using such objective measures of sleep quality to further investigate its contribution to motor learning in PD.

### 4.2. Sensorimotor Network Segregation as a Neural Signature of Motor Learning Capacity

We showed that sensorimotor network segregation was a strong predictor of the overall gains in writing accuracy after training. Notably, this result supports findings from young adults showing higher sensorimotor within-network RS-FC [28] and lower sensorimotor-visual between-network RS-FC [20] being predictive of better motor learning. Although portraying a less-connected sensorimotor network as being more efficient seems counterintuitive, it likely reflects a larger capacity for reconfiguration into a more integrated state based on task demands [65]. Indeed, Monteiro et al. (2020), showed that as task complexity of a bimanual coordination task increased, the sensorimotor and visual networks became more connected with each other and more segregated from other task-irrelevant networks in young adults. Older adults, however, who had less segregated networks at rest [26], were unable to similarly modulate their connectivity with increasing task complexity, which reflected in worse task performance [27].

Previously, we demonstrated in the same cohort that connectivity within the DAN increased significantly from pre- to post-training, which we interpreted as cognitive compensatory involvement responsible for robust learning [25]. Within the framework of segregation, increased within-network connectivity also translates to more segregated processing. Several studies have also shown more segregated brain networks as a result of motor learning in healthy individuals [66,67,68], the specific networks involved varying in relation to the particular training mode [68] or type of feedback provided [67]. Here, we did not find DAN segregation to be predictive of motor learning capacity, suggesting that changes in the DAN are the result of the specific training provided, while the efficiency of the sensorimotor network may determine the ability to flexibly explore and respond to progressive training demands, and thereby predict motor learning capacity.

Finally, sensorimotor network segregation as defined in the present study is a rather coarse metric, not specific to the visual–sensorimotor nor to the attentional–sensorimotor interactions, which are more related to the writing task [34]. This lack of specificity, however, makes sensorimotor network segregation a promising marker for characterizing motor learning capacity for a variety of motor tasks in PD. Measuring network segregation at rest also increases feasibility for paradigms that are challenging to perform in the scanner, or that may result in highly variable execution. Future work should seek to validate this marker with larger PD cohorts and varied motor learning tasks.

### 4.3. Two to Tango—Preserved System Hardware and the Room to Improve

The results of this study are largely in line with previous work in PD showing that lower baseline values and more preserved “system hardware” are predictive of larger learning gains [18,19,55,56,59]. Lower baseline values indicate lower than average performance relative to the rest of the sample on a particular task, which we interpret as being undertrained on that particular task, and thereby having a larger room to improve. Conceptually, the utility of lower baseline values as a predictor of learning may be impacted by inadequate sample variability as well as task context. For instance, ceiling effects or floor effects on very simple or challenging tasks may reduce sample variability and its predictive utility. Further, it is also unclear whether baseline values would be predictive of learning gains when the training target increases adaptively as a function of performance. Nevertheless, poor baseline performance should not discourage clinicians and patients to implement motor task training, as our results showed more potential for improvements.

By system hardware, we adopt the terminology of Ophey and colleagues [59] who use it to refer to the resources “to acquire, implement and sharpen” learning strategies, based on the compensation and magnification model [69]. In different learning contexts, the precise piece of hardware that is important differs, such as balance confidence for highly challenging balance training [18], years of education and fluid intelligence for working memory training [59], visual episodic memory for bilateral motor sequence learning [70], global cognition for dual task gait training [19], and perhaps female sex for language-related tasks such as in this study.

Limited work has been carried out on the neural hardware of motor learning capacity, with one study showing that greater cortical thickness of the visual, dorsal attention, frontoparietal and lateral sensorimotor networks predicted larger improvements in dual task gait speed subsequent to 6 weeks of cognitively challenging gait and balance training [55]. Although freezers and non-freezers showed slightly different network associations, these findings further support the need for a multi-network approach when investigating markers of motor learning capacity in PD.

### 4.4. Clinical Implications and Limitations

This study is one of the few addressing the following question: “who is likely to benefit most from motor learning?”, with the ultimate aim to achieve personalized rehabilitation for people with PD. Given the effort required for training, it is imperative to better understand patients’ training potential at rehabilitation onset. This study shows that adding a short 5 min RS-fMRI scan could aid in the prediction when combined with unimanual dexterity and disease severity measures (disease duration, motor symptom severity and FOG presence) that are already common clinical tests. Also, initial writing performance proved important as an indicator for the ‘room for improvement’, illustrating the relevance of using sufficiently challenging tasks to measure improvements. Still, the current results must be interpreted with caution as they are based on a limited and convenient sample and likely to be overoptimistic. In addition, a liberal threshold of *p* < 0.1 was used for retaining predictors in the regression model. Even though the risk of overfitting was counteracted with rigorous bootstrapping and PCA methods, future studies are needed to replicate the findings in different cohorts, including a control group of healthy subjects, while addressing different task training. Our study sample only included patients with intact cognitive function, showing no impact of cognition on training success. Nevertheless, future studies should include patients with cognitive dysfunctions, as up to 80% of patients develop dementia over the disease course [71]. Moreover, global cognitive assessments may not be sufficient to predict motor learning capacity in PD, as different cognitive subdomains are affected, such as executive function and memory, which likely impact more seriously on learning [7,72].

## 5. Conclusions

Motor learning potential is progressively affected in Parkinson’s disease, which presents a clinical challenge to interventions aiming to teach patients new skills or relearn previously learned skills, such as handwriting. Clinical and neural markers of motor learning potential may facilitate clinical decision-making regarding the optimal conditions for undertaking such motor learning approaches. In this study, we therefore investigated clinical measures as well as functional brain organization as markers of motor learning potential following writing training in PD. The results showed that baseline writing performance, female sex, and resting-state sensorimotor network segregation predict long-term motor learning capacity following writing training in people with PD. Though requiring further validation, these findings provide a promising marker to predict motor learning potential and to personalize rehabilitation in PD.

## Figures and Tables

**Figure 1 brainsci-14-00376-f001:**
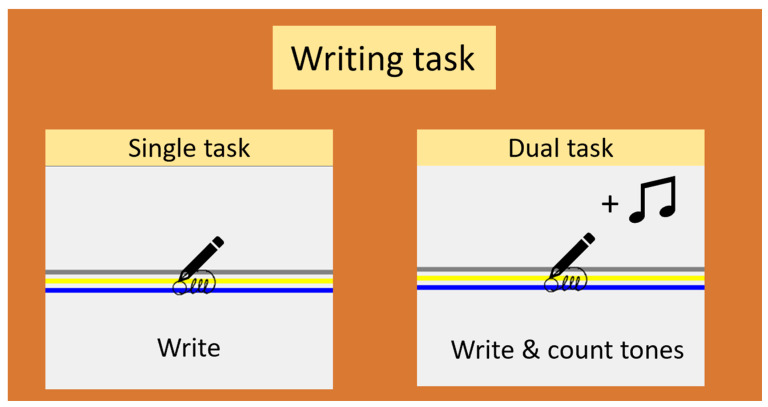
Tasks used to assess learning. The writing task consisted of a three-loop sequence written between target lines, either alone, or in combination with a cognitive dual task. Upon completion of the three loops, participants had to return to the starting point to initiate a new sequence. Writing performance was assessed by amplitude, amplitude variability, deviation from target, and average speed.

**Figure 2 brainsci-14-00376-f002:**
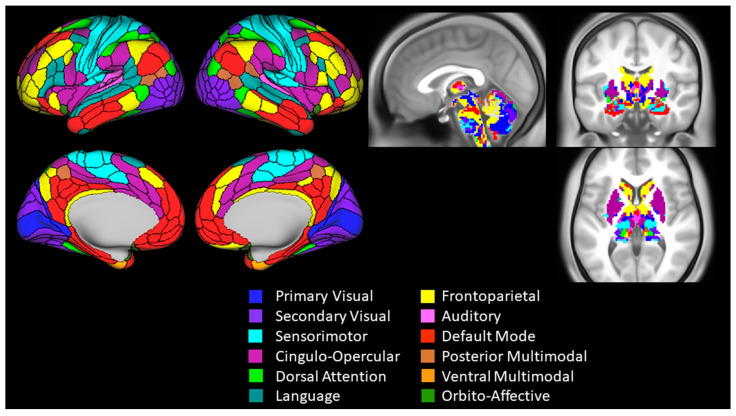
Cole-Anticevic Brain-wide Network Partition. Cortical parcel (**left**) and subcortical voxel (**right**) assignments to the 12 networks are shown on the Human Connectome Project average surface and volume templates (S1200 release), respectively.

**Figure 3 brainsci-14-00376-f003:**
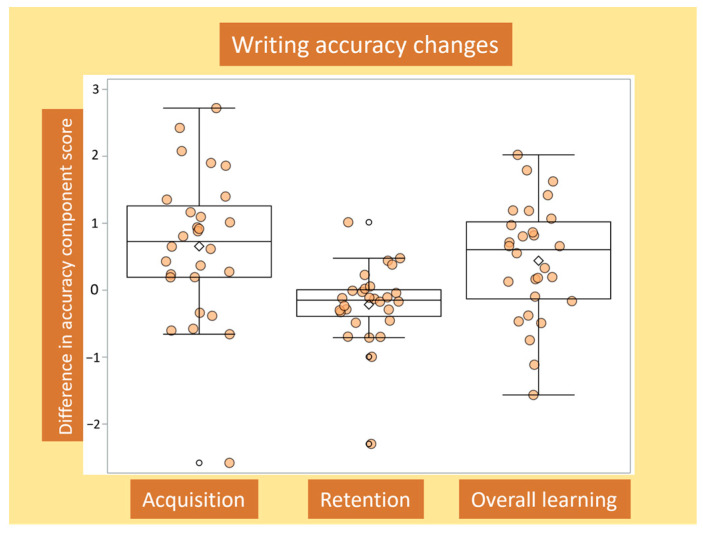
Writing accuracy changes over time. Change in the accuracy component score, which was combined for single and dual tasks, for acquisition, retention, and overall learning shown for each participant (filled circles) and for the whole group (box plots). Means are indicated by unfilled diamonds, and mild outliers by unfilled circles.

**Figure 4 brainsci-14-00376-f004:**
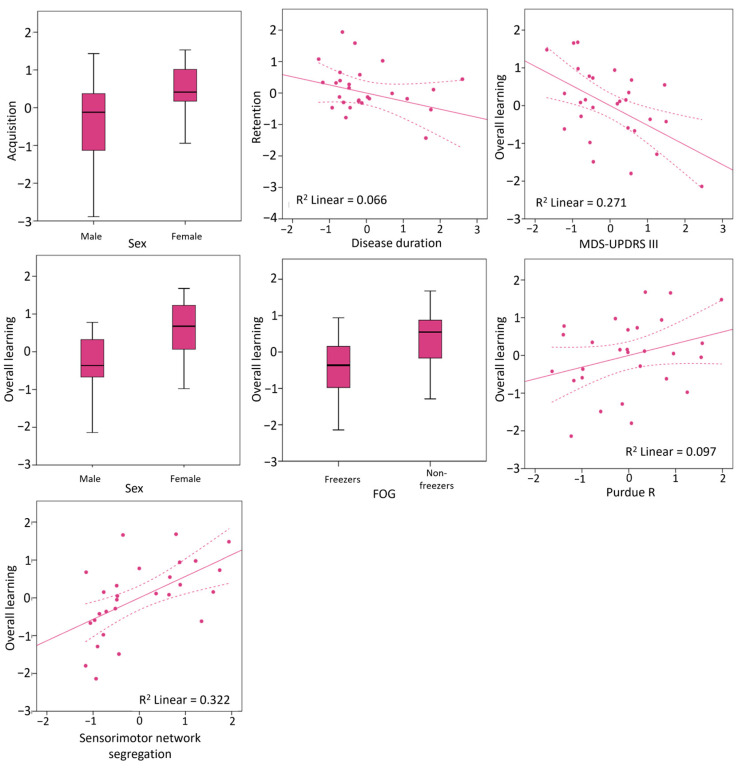
Box and scatter plots showing the associations between baseline clinical and neural predictors and writing accuracy gains after accounting for the contributions of baseline writing accuracy (residuals visualized). Scatter plots also depict the linear line of best fit (solid line) and its 95% confidence intervals (dashed lines).

**Table 1 brainsci-14-00376-t001:** Clinical demographics.

Measure (Units)	Mean (SD)	Range
Age (years)	63.93 ± 8.58	46–78
Sex (M/F)	17/11	
EHI (%)	100 (80; 100)	7.7–100
H&Y (1–5)	2 (2; 2)	1–4
Disease duration (years)	6.89 ± 3.93	1–17
FOG presence (Yes/No)	13/15	
LEDD (mg/24 h)	641.5 ± 288.47	126–1417.5
MDS-UPDRS-III (0–132)	31.14 ± 15.07	6–63
Sleep complaints (0–8)	3.71 ± 1.71	0–7
Purdue Pegboard Right	8.64 ± 2.71	3–14
MMSE (0–30)	29 (28; 29)	25–30
MoCA (0–30)	26.54 ± 1.73	22–29
HADS-anxiety (0–21)	6.32 ± 4.16	0–14
HADS-depression (0–21)	5.29 ± 3.21	0–13

Mean and standard deviations or medians with first and third quartiles (H&Y and MMSE) along with the ranges across the sample are shown here. SD—standard deviation, M—male, F—female, EHI—Edinburgh Handedness Inventory, H&Y—Hoehn and Yahr, FOG—freezing of gait, LEDD—Levodopa equivalent dose, MDS-UPDRS-III—Movement Disorders Society sponsored revisions of the Unified Parkinson’s Disease Rating Scale part III, MMSE—Mini Mental Status Examination, MoCA—Montreal Cognitive Assessment; HADS—Hospital Anxiety and Depression Scale.

**Table 2 brainsci-14-00376-t002:** Associations between baseline predictors and writing accuracy gains, accounting for baseline writing accuracy.

	Acquisition (Pre to Post-Training)				Retention (Post-Training to Follow-Up)				Overall Learning (Pre-Training to Follow-Up)			
	R	*p*	BCa 95%CI		R	*p*	BCa 95%CI		R	*p*	BCa 95%CI	
			Lower	Upper			Lower	Upper			Lower	Upper
**Clinical measures**												
Age	0.00	0.985	−0.45	0.55	−0.16	0.433	−0.55	0.27	−0.10	0.611	−0.45	0.30
Female sex	**0.43**	**0.024**	**0.10**	**0.69**	−0.02	0.935	−0.40	0.26	**0.51**	**0.007**	**0.20**	**0.74**
LEDD	0.13	0.523	−0.23	0.47	−0.18	0.377	−0.61	0.20	0.03	0.868	−0.39	0.59
Non-freezer	0.26	0.183	−0.15	0.61	0.19	0.351	−0.26	0.51	0.44	0.021	0.06	0.74
Disease duration	−0.02	0.930	−0.43	0.40	−0.26	0.195	−0.53	−0.01	−0.20	0.326	−0.60	0.22
MDS-UPDRS III	*−0.45*	*0.020*	*−0.82*	*0.23*	0.02	0.923	−0.39	0.40	−0.52	0.005	−0.79	−0.08
Sleep complaints	0.23	0.245	−0.28	0.59	*−0.40*	*0.038*	*−0.72*	*0.18*	0.00	0.982	−0.45	0.39
MoCA	0.13	0.525	−0.25	0.55	−0.02	0.912	−0.23	0.14	0.14	0.493	−0.28	0.56
HADS-anxiety	0.04	0.841	−0.48	0.52	−0.09	0.640	−0.47	0.32	−0.02	0.939	−0.52	0.46
HADS-depression	−0.06	0.782	−0.40	0.35	−0.08	0.678	−0.35	0.23	−0.12	0.539	−0.48	0.26
Purdue Unimanual Right	0.21	0.306	−0.20	0.48	0.10	0.628	−0.47	0.55	0.31	0.114	0.02	0.56
**Network Segregation**												
Primary visual	0.00	0.986	−0.52	0.51	0.13	0.523	−0.35	0.45	0.08	0.680	−0.42	0.53
Secondary visual	0.01	0.964	−0.47	0.35	−0.06	0.755	−0.41	0.32	−0.03	0.875	−0.41	0.29
Sensorimotor	0.31	0.111	−0.08	0.60	0.28	0.156	−0.11	0.62	0.57	0.002	0.29	0.78
Cingulo-opercular	−0.16	0.423	−0.47	0.18	0.15	0.449	−0.22	0.51	−0.09	0.658	−0.39	0.23
Dorsal attention	0.27	0.178	−0.11	0.58	−0.09	0.660	−0.29	0.16	0.26	0.191	−0.15	0.55
Language	−0.20	0.306	−0.49	0.18	0.02	0.910	−0.26	0.37	−0.23	0.249	−0.49	0.12
Frontoparietal	−0.06	0.752	−0.49	0.26	0.11	0.579	−0.25	0.44	0.00	0.998	−0.46	0.40
Auditory	0.23	0.252	−0.22	0.56	−0.10	0.607	−0.50	0.41	0.20	0.310	−0.21	0.50
Default	0.00	0.988	−0.37	0.39	0.09	0.651	−0.20	0.36	0.06	0.772	−0.32	0.48
Posterior multimodal	0.16	0.423	−0.12	0.41	−0.18	0.369	−0.48	0.08	0.07	0.727	−0.33	0.43
Ventral multimodal	−0.24	0.236	−0.52	0.26	0.24	0.230	−0.33	0.57	−0.12	0.550	−0.46	0.37
Orbito-affective	−0.11	0.576	−0.49	0.24	−0.14	0.500	−0.51	0.42	−0.23	0.254	−0.59	0.15

Inference was made based on the confidence interval not including 0. Significant predictors are highlighted in bold. Note that some predictors showed significant associations in the original sample (shown in italic), but their bootstrap confidence intervals included 0, and vice versa. R—Partial correlation in original sample, *p*—*p*-value for partial correlation in original sample, BCa 95%CI—Bias-corrected and accelerated bootstrap 95% confidence intervals. LEDD—Levodopa equivalent dose, MDS-UPDRS-III—Movement Disorders Society sponsored revisions of the Unified Parkinson’s Disease Rating Scale part III, MoCA—Montreal Cognitive Assessment; HADS—Hospital Anxiety and Depression Scale.

## Data Availability

The data presented in this study are available on request from the corresponding author, subject to privacy, ethical and legal clearance.

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
