# Peer review of "Sensorimotor Network Segregation Predicts Long-Term Learning of Writing Skills in Parkinson’s Disease"

_brainsci, 2024, doi:10.3390/brainsci14040376_

Round 1

Reviewer 1 Report

Comments and Suggestions for Authors

All proposals for rehabilitation in neurodegenerative diseases are valid, but I would like to know more about the scientific basis for this type of approach in addition to the group's experience published in previous data. I think it would be more impartial if the introduction was also based on other studies that are not from the group, because it refers to the same sample as the current work. Furthermore, I suggest that the type of motor training and why this therapy time is chosen be introduced in more detail. So, what are the benefits of micrographia training for Parkinson's disease?

Regarding the methodology, how did the researchers evaluate the effective implementation of the proposed training? What guarantee did the researcher have that the training was being carried out correctly?

Why in the experimental procedure item was a task described to perform a functional task examination and in the acquisition of neuroimaging data was it described to perform a functional examination in the resting state? The methodology is very confusing. It would be interesting for the authors to describe only the data they used in this study and clearly state what was used from previous studies.

Does item 2.4 refer to an analysis of data collected during a task-fMRI or before performing the rs-fMRI? Part B of Figure 1 seems like a result to me, therefore it should not be exposed in the methodology session.

Regarding the results, it would be interesting to show the clinical data by groups, PD with FOG and non-FOG in table 1 and also include a statistical analysis of demographic metrics, as shown in previous articles.

I suggest including Table S1 in the body of the manuscript and not as supplementary material and the same for Figure S2. Why was it included as supplementary material?

I was unable to understand when the connectivity data in items 3.1 and 3.2 of the result was used. I suggest that the authors align the methodological description with the results. The study proposal and study outcomes are very confusing.

Comments on the Quality of English Language

Some typing and formatting errors

Reviewer 2 Report

Comments and Suggestions for Authors

This is a fascinating and valuable study.  As the authors point out, the fact that *less* integration of different networks predicts *better* learning performance is not intuitively obvious, and having a measure that correlates with potential for remediation may contribute significantly to prognosis and resource utilization.

The three outcomes studied were acquisition, retention, and overall learning which is acquisition plus retention.  Thus the third outcome is not independent of the first two; in such a case it would be valuable to look at the variation in degree to which each of the first two contributed to the third. Moreover, the fact that a lower starting baseline performance predicts greater degree of improvement is an important finding that should be further discussed.  It is not necessarily "expected" (Line 310). A trivial explanation would be that such patients simply have more room to improve; but the fact that poor baseline does not imply lesser potential is very, very important in clinical decision making.

The unsupervised nature of the training is important and differs from most such studies.  The fact that there was no correlation with HADS or MOCA measures is important, and should also be further emphasized (Line 329), as a trivial explanation for the result would be lack of motivation or cognitive problems with an unsupervised program. That said, it is a little surprising that retention correlated negatively with disease duration if MOCA showed no cognitive issues.  What other than cognitive problems would explain this?

Line 36: many of the non-motor symptoms are not dopamine responsive, e.g. cognitive decline, which would be of direct relevance to a study involving writing; this is a minor point for an analysis like this but has implications for treatment. For example, cholinergic centers in the forebrain and brainstem are also affected.

Line 42: The study is oriented around the idea of "rehabilitation," a concept that makes more sense when applied to the result of a one-event process such as head injury or stroke.  Parkinson's is a progressive disorder.  I would advise referring to the training here just as "therapy," whether speech, cognitive, or occupational, rather than "rehabilitation."  Patients may improve with therapy but the disease is progressive. Gains made through therapy are likely to be lost over time (even given the retention measured here) and it is important to emphasize that strategies based on the findings of studies like this would need to be implemented in an ongoing fashion.

Reviewer 3 Report

Comments and Suggestions for Authors

An interesting manuscript on the factors influencing long-term learning of writing skills in 28 pacientes with PD, including the presence of freezing of gait in nearly half of them. Absence of freezing of gait, female sex, less severe motor symptoms and better sensorimotor network segragation seem to be associated with better long-term learning or writting skills. I suggest as minor revision to discuss the strengths and limitations of the study with more detail.  

Reviewer 4 Report

Comments and Suggestions for Authors

Dear Authors,

-The abstract could benefit from adding more information about how the study was performed;

- The keyword are suitable for the manuscript's subject;

-I feel that the introduction contains not enough information in order to give the amount necessary to the readers in order to continue to read your article;

- The material and methods are described with enough information and insight;

-The results are very interesting and complete;

- The discussion are very well written and bring a lot of valuable metrics;

- I would add the limitations in a separate chapter;

- The conclusion lacks substance and more information should be added;

- The references should bring more articles in the manuscript. 

Overall is an interesting work. Congrats!

Round 2

Reviewer 1 Report

Comments and Suggestions for Authors

The authors understood the need for all the changes mentioned and carried them out appropriately so that the article became clearer, more objective and understandable regarding the objective, method and result of the study. Therefore, I believe that as presented, the manuscript is suitable for publication.